# Proteome Profiling of Rabies-Infected and Uninfected Dog Brain Tissues, Cerebrospinal Fluids and Serum Samples

**DOI:** 10.3390/proteomes13040066

**Published:** 2025-12-15

**Authors:** Ukamaka U. Eze, Rethabile Mokoena, Kenneth I. Ogbu, Sinegugu Dubazana, Ernest C. Ngoepe, Mparamoto Munangatire, Romanus C. Ezeokonkwo, Boniface M. Anene, Sindisiwe G. Buthelezi, Claude T. Sabeta

**Affiliations:** 1Department of Veterinary Medicine, Faculty of Veterinary Medicine, University of Nigeria, Nsukka 410001, Nigeria; boniface.anene@unn.edu.ng; 2Future Production, Chemicals, Council for Scientific and Industrial Research, Pretoria 0001, South Africa; rmokoena1@csir.co.za (R.M.); snedubazana@gmail.com (S.D.); SButhelezi@csir.co.za (S.G.B.); 3Department of Animal Health, National Veterinary Research Institute, Federal College of Animal Health and Production Technology, P.M.B. 05, Vom 930001, Nigeria; drken2016@gmail.com; 4Agricultural Research Council, Onderstepoort Veterinary Research, Pretoria 0110, South Africa; ngoepee@arc.agric.za (E.C.N.); mparamotom@arc.agric.za (M.M.); 5Department of Veterinary Parasitology and Entomology, Faculty of Veterinary Medicine, University of Nigeria, Nsukka 410001, Nigeria; romanus.ezeokonkwo@unn.edu.ng; 6Department of Veterinary Tropical Diseases, Faculty of Veterinary Science, University of Pretoria, Pretoria 0110, South Africa; claude.sabeta@up.ac.za

**Keywords:** rabies, proteome profiling, antemortem diagnosis, dogs

## Abstract

Background: Rabies is among the oldest known zoonotic viral diseases and is caused by members of the *Lyssavirus* genus. The prototype species, *Lyssavirus rabies*, effectively evades the host immune response, allowing the infection to progress unnoticed until the onset of clinical signs. At this stage, the disease is irreversible and invariably fatal, with definitive diagnosis possible only post-mortem. Given the advances in modern proteomics, this study aimed to identify potential protein biomarkers for antemortem diagnosis of rabies in dogs, which are the principal reservoir hosts of the rabies virus. Methods: Two hundred and thirty-one samples (brain tissues (BT), cerebrospinal fluids (CSF), and serum (SR) samples) were collected from apparently healthy dogs brought for slaughter for human consumption in South-East and North-Central Nigeria. All the BT were subjected to a direct fluorescent antibody test to confirm the presence of lyssavirus antigen, and 8.7% (*n* = 20) were positive. Protein extraction, quantification, reduction, and alkylation were followed by on-bead (HILIC) cleanup and tryptic digestion. The resulting peptides from each sample were injected into the Evosep One LC system, coupled to the timsTOF HT MS, using the standard dia-PASEF short gradient data acquisition method. Data was processed using Spectronaut^TM^ (v19). An unpaired *t*-test was performed to compare identified protein groups (proteins and their isoforms) between the rabies-infected and uninfected BT, CSF, and SR samples. Results: The study yielded 54 significantly differentially abundant proteins for the BT group, 299 for the CSF group, and 280 for the SR group. Forty-five overlapping differentially abundant proteins were identified between CSF and SR, one between BT and CSF, and two between BT and SR; none were found that overlapped all three groups. Within the BT group, 33 proteins showed increased abundance, while 21 showed decreased abundance in the rabies-positive samples. In the CSF group, 159 proteins had increased abundance and 140 had decreased abundance in the rabies-positive samples. For the SR group, 215 proteins showed increased abundance, and 65 showed decreased abundance in the rabies-positive samples. Functional enrichment analysis revealed that pathways associated with CSF, spinocerebellar ataxia, and neurodegeneration were among the significant findings. Conclusion: This study identified canonical proteins in CSF and SR that serve as candidate biomarkers for rabies infection, offering insights into neuronal dysfunction and potential tools for early diagnosis.

## 1. Introduction

Rabies is a long-standing, neglected, and invariably fatal zoonotic disease that affects all mammalian species, including humans [1,2]. The aetiologic agent primarily targets the central nervous system and progresses to an encephalitis, which has a case fatality rate of 100% once clinical symptoms appear [3]. In Africa and Asia, an estimated 95% of human rabies cases are attributed to domestic dogs, resulting in approximately 60,000 human deaths annually [4]. The causative agent of rabies is *Lyssavirus rabies* (RABV), one of 18 recognized species within the *Lyssavirus* genus [5]. Despite the high fatality associated with the infection, rabies is entirely preventable through the regular administration of anti-rabies vaccines to dogs. Achieving a minimum vaccination coverage of 70% of the canine population has been demonstrated as the most effective strategy for eliminating rabies in endemic areas because dogs are the principal reservoir and source of rabies [6,7]. However, in many low- and middle-income countries, reaching this threshold remains a challenge due to resource constraints, limited veterinary workforce, and lack of political commitment [8]. These barriers contribute to the continued endemicity of rabies in these regions, thereby sustaining the risk of dog-mediated transmission to humans.

Post-exposure prophylaxis (PEP) is recommended for individuals exposed to suspected or confirmed rabid dog bites [7]. However, the high cost of PEP and its limited availability, particularly in rural and resource-constrained settings, pose significant challenges to timely and equitable access. To optimize the use of PEP and minimize unnecessary expenditures, it is essential to clinically differentiate between exposures caused by rabid and non-rabid animals. Unfortunately, the absence of a reliable antemortem diagnostic test for rabies hampers such clinical decision-making [9], often resulting in either under-treatment or the unnecessary administration of PEP.

The clinical progression of rabies is typically divided into distinct phases. The initial phase, known as the prodromal stage, is characterized by non-specific symptoms such as fever, malaise, and behavioural changes. This is followed by a latent or incubation period, during which the virus remains asymptomatic as it travels through the peripheral nervous system toward the central nervous system. Clinical signs during these early stages are often indistinguishable from other febrile or neurological illnesses, making early diagnosis challenging. Once the virus reaches the brain, it manifests as either the furious (excitative) form, characterized by hyperactivity, aggression, and hydrophobia or the paralytic (dumb) form, which presents with flaccid paralysis, progressive neurological dysfunction, coma and death [10,11].

The RABV evades the host immune system, making the disease detectable only upon the onset of clinical symptoms, at which point it is invariably fatal [12]. Due to its ability to evade the host immune response, rabies is typically diagnosed post-mortem, at which point therapeutic intervention is no longer possible.

To this end, we employed a proteomics approach to identify potential candidate markers from brain tissues (BT), cerebrospinal fluid (CSF), and sera (SR) collected from both rabies-positive and rabies-negative dogs slaughtered for human consumption. Proteomics has been widely applied in the identification of biomarkers for the diagnosis of various diseases, including neurodegenerative disorders, metabolic syndromes, and different forms of cancer from tissues and body fluids [13,14,15,16]. It has provided deep insights into the molecular mechanisms of diseases and cellular functions [17,18]. Previous proteomics studies on the BT of rabid dogs and humans identified several promising candidate markers, such as KPNA4, CAMK2A, ApoD and RAC1 proteins involved in diverse molecular pathways associated with rabies pathogenesis [19,20,21,22,23,24]. However, these markers were identified in animals that had succumbed to the disease. While such proteins may represent potential diagnostic targets, their utility as early biomarkers remains uncertain, as the proteome profile during the initial stages of infection may differ significantly from that observed at terminal stages of the disease.

This study aimed to identify differentially abundant canonical proteins in rabies-positive and rabies-negative dogs that can function as potential markers during the early stages of rabies infection and may serve as targets for antemortem diagnosis before the onset of clinical symptoms, thereby facilitating early clinical diagnosis.

## 2. Materials and Methods

### 2.1. Ethical Approval

This study was conducted in adherence to the Faculty of Veterinary Medicine, University of Nigeria, Nsukka’s guidelines for animal husbandry, which corresponds with the National Institute of Health (NIH) guidelines [25]. All protocols for this study were reviewed and approved on 28 March 2023 by the Institutional Animal Care and Use Committee (IACUC) of the Faculty of Veterinary Medicine, University of Nigeria, Nsukka (Approval No: FVM-UNN-IACUC-2023-03100). BT, CSF and SR were obtained with informed consent from dog traders and dog meat vendors.

Additionally, the ethical approval for the use of animals in research was granted on 2 September 2024 by the South African National Animal Ethics Committee in accordance with Section 20 of the Animal Diseases Act of 1984 (Act No. 35 of 1984), reference number 12/11/1/1/8 (6640KL). This dual ethical approval underscores the adherence of this study to internationally recognized principles of animal welfare and scientific integrity.

### 2.2. Study Area and Sample Size

The samples were collected in the North-Central and South-East geopolitical zones of Nigeria. These regions were purposely selected due to the high concentration of dog markets. One state from each region was selected through a simple random sampling technique—Plateau State (North-Central) and Enugu State (South-East). Within each state, two major dog markets known for the slaughter of dogs for human consumption were purposely chosen as sampling sites.

The required sample size was calculated using the formula by Thrusfield et al. [26], based on a previously reported prevalence rate of 8.3% for rabies in apparently healthy dogs in South-East Nigeria [27]. This yielded a minimum sample size of 117 dogs. However, to improve the statistical power and representativeness of the study, a total of 231 samples were collected.

### 2.3. Exclusion Criteria

Dogs exhibiting clinical signs suggestive of illness, including but not limited to dullness, excessive salivation (drooling), emaciation, ocular or nasal discharges, diarrhoea, and vomiting, were excluded from the study. Additionally, dogs that had already been slaughtered before the researchers arrived at the market were excluded from the sampling.

### 2.4. Sample Collection

A total of 231 samples, each of BT, SR and CSF, were collected from dogs slaughtered in selected markets across North-Central and South-Eastern Nigeria. The dogs originated from household settings within these regions of Nigeria, where their rearing is partly driven by the cultural practice of dog meat consumption, in which dog meat is regarded as a delicacy. Each sample set was labelled using the corresponding identification tag assigned to each dog at the time of sampling. All collected samples were transported under cold chain conditions to the Rabies Unit of the Central Diagnostic Laboratory at the National Veterinary Research Institute (NVRI), Vom, Plateau State, Nigeria, for lyssavirus antigen detection.

Detection of lyssavirus antigen was performed using the Direct Fluorescent Antibody (DFA) test following the standard protocol described by Dean et al. [28]. Of the 231 brain tissue samples tested, 8.7% (n = 20) were confirmed positive for lyssavirus antigen. The corresponding CSF and SR samples from these positive cases were also categorized as lyssavirus-positive.

For comparative purposes, 20 lyssavirus-negative samples were randomly selected from each sample type (BT CSF, and SR) to serve as controls. All samples were subsequently stored at −20 °C pending further analyses.

### 2.5. Protein Extraction

A total of 40 experimental replicates were analyzed for each of the BT, CSF, and SR sample types, comprising 20 positive and 20 negative samples. Protein extraction from BT was carried out using the TRIzol™ reagent method according to the manufacturer’s instructions (Thermo Fisher Scientific, Waltham, MA, USA). For CSF and SR, proteins were extracted using the acetone precipitation method as described by Nejadi et al. [29], with slight modifications.

Briefly, 100% acetone and ethanol were pre-chilled at −20 °C and 4 °C, respectively. For each sample, four volumes (200 µL) of pre-chilled acetone were added to one volume (50 µL) of either SR or CSF in protein LoBind tubes. The mixture was vortexed for 30 s and incubated at −20 °C for 1 h. Following incubation, the samples were centrifuged at 14,000× *g* for 10 min at 4 °C. The resulting supernatants were carefully discarded, and the protein pellets were washed with ice-cold ethanol. After removing the ethanol, the pellets were air-dried for approximately one minute.

The dried pellets were then re-suspended in 200 µL of 2% sodium dodecyl sulfate (SDS) in 50 mM Tris-HCl (pH 8.0), heated at 90 °C for 10 min in a heat block, and vortexed vigorously for 30 s until completely solubilized. The resulting protein extracts were stored at −80 °C until further analysis.

### 2.6. Protein Quantification, Digestion, and LC-MS/MS Analysis

Protein concentrations in CSF, BT, and SR samples were determined using the bicinchoninic acid (BCA) assay as per the manufacturer’s instructions. Following quantification, only samples containing at least 20 µg of protein were selected for further analysis. As a result, 10 rabies-infected and -uninfected samples from each sample type were utilized for sample preparation, where 20 µg of total protein from each sample was reduced with dithiothreitol (DTT; 10 mM, 30 min at 60 °C), then alkylated using iodoacetamide (IAA; 40 mM, 30 min at room temperature in the dark), and subsequently quenched with 10 mM DTT. Sample clean-up and digestion of proteins were performed on-bead using MagReSyn^®^ HILIC microspheres (ReSyn Biosciences, GP, Pretoria, South Africa). Sample clean-up was performed using hydrophilic interaction liquid chromatography (HILIC) procedure as outlined by Baichan et al. [30], and tryptic digestion of proteins was performed at a 1:20 (enzyme: protein) ratio for 2 h at 47 °C. The resulting peptides were stored at −80 °C and subsequently dried using a vacuum concentrator (CentriVap^®^ Concentrator, Labconco, Kansas City, MO, USA).

Before bottom-up LC-MS/MS analysis, peptides were reconstituted in 2% acetonitrile (ACN) containing 0.2% formic acid (FA). Peptide concentrations were determined using the Pierce Quantitative Colorimetric Peptide Assay, following the manufacturer’s protocol. Thereafter, 0.5 µg of peptides per sample were loaded onto Evotips following the manufacturer’s instructions. Samples were analyzed on the Evosep One LC system (EV-1000), coupled to a timsTOF HT mass spectrometer (Bruker Daltonics, Billerica, MA, USA) operating in positive ion mode with the CaptiveSpray ion source.

Peptide separation was performed using a C18 Evosep Performance column (8 cm, 1.5 µm × 150 µm; EV-1109) (Evosep, Odense, Denmark) with the column oven set at 40 °C, at a flow rate of 1 µL/min using the standard 60 samples per day (60SPD) method. The standard 60 SPD method had peptides eluted over a 21 min gradient consisting of a preformed gradient of 5–30% Solvent B and an offset gradient with a lower percentage of ACN using Solvent A (Solvent A: LC-MS grade water with 0.1% FA; Solvent B: 100% LC-MS grade ACN with 0.1% FA) (Thermo Fisher Scientific^TM^, Waltham, MA, USA). Data was acquired using the standard dia-PASEF short gradient method, across a mass range of 100–1700 *m*/*z* and an ion mobility range of 0.85–1.30 V·s/cm^2^ with 21 ion mobility windows. The source parameters were as follows: capillary voltage was set to 1600 V, the dry gas was set to 3.0 L/min, and the dry temperature was 180 °C. The ramp and accumulation time was 100 milliseconds, the ramp rate was 9.42 Hz, and the MS1 and MS2 cycle times were set at 0.958 s. The method used a duty cycle of 100%.

### 2.7. Data Analysis

Raw mass spectrometric data was processed using Spectronaut™ 19 (Biognosys, Schlieren, Switzerland), employing the directDIA + workflow. Default settings were applied, with enzyme specificity set to trypsin/P, allowing up to two missed cleavages. Fixed modification was carbamidomethylation of cysteine, and variable modifications included N-terminal acetylation and methionine oxidation. Protein and peptide identifications were filtered using a 1% false discovery rate (FDR), with a minimum peptide length of 7 amino acids, and peptides were filtered by selecting the best-scoring fragment ions per precursor ion, with a minimum of 3 and a maximum of 6 fragment ions per precursor ion. Label-free quantification (LFQ) was conducted based on area under the curve (AUC) of fragment ion intensities, without imputation. Protein inference was carried out using the IDPicker algorithm, with protein identification based on a Uniprot-reviewed *canis lupus familiaris* FASTA protein database, including common contaminants [31], and a minimum of 2 peptides required for protein inference. Samples were grouped into positive and negative categories based on their lyssavirus status.

### 2.8. Retrospective Power Analysis

A power analysis was performed for the three sample types (BT, CSF, and SR) using MSstats (v3.4.1) [32] in RStudio (v2022.12.0 + 353) [33]. The power analysis was conducted to determine a desired fold-change (or effect size) to establish the significance criteria for selecting differentially abundant proteins for this study. The retrospective power analysis was performed at a power of 80%, with an FDR of 5%. A sample size of 20 patient samples was used to perform the analysis.

### 2.9. Functional Enrichment Analysis

ShinyGo 0.81 (https://bioinformatics.sdstate.edu/go/ accessed on 23 February 2025) was used to evaluate functional enrichment of the significantly differentially abundant proteins identified in each group (BT, CSF, and SR) at an FDR of 5% using the KEGG pathways database. The species was set to *canis lupus familiaris*. The full list of identified protein groups in Spectronaut^TM^ 19 for each group was used as the recommended background list. The number of pathways shown was set to 20, with a pathway size of a minimum of 2 and a maximum of 5000, and the ‘remove redundancy’ setting was selected.

## 3. Results

### 3.1. Proteome Profiling Overview

LC-MS/MS analysis was conducted on three different sample sources (BT, CSF and SR) consisting of lyssavirus-positive and negative samples. Overall, an average of 6672 and 6547 protein groups (proteins and their isoforms), and an average of 52,597 and 50,668 modified peptides (peptide sequences with modifications of amino acids) were identified in the rabies-positive (BT positive) and negative (BT negative) brain tissue samples, respectively. For the CSF samples, an average of 1362 and 1368 protein groups, as well as an average of 8069 and 7542 modified peptides were identified in rabies-positive (CSF positive) and -negative (CSF negative) samples, respectively. An average of 1455 and 1500 protein groups, and an average of 8396 and 8373 modified peptides were identified in the rabies-positive (SR positive) and -negative (SR negative) serum samples, respectively (Figure 1).

In the BT group, a total of 54 differentially abundant proteins were identified as significant following an unpaired Student’s *t*-test; 33 showed increased abundance, while 21 showed decreased abundance, as illustrated in Figure 2. In the CSF group, a total of 299 differentially abundant proteins were identified as significant following an unpaired Student’s *t*-test; 159 had increased abundance, while 140 had decreased abundance in the positive group relative to the negative group, as illustrated in Figure 3. Within the SR group, a total of 280 differentially abundant proteins were identified as significant following an unpaired Student’s *t*-test; 215 showed decreased abundance, while 65 showed decreased abundance in the positive group compared to the negative group, as shown in Figure 4.

### 3.2. Overlapping Proteins Between the Different Sample Types

A Venn diagram was created to identify overlapping proteins that were found to be significantly differentially abundant between the rabies-positive and -negative samples of the BT, CSF, and SR groups. The comparison revealed: 45 proteins were found between the CSF and SR groups, 1 protein was shared between the BT and CSF groups, and 2 proteins were common to the BT and SR groups. However, no proteins were found to overlap among all three groups as shown in Figure 5.

The candidate protein marker that overlapped between CSF and BT is Rho GDP dissociation inhibitor beta (ARHGDIB). The two (2) candidate protein markers that overlapped between BT and SR are:Myelin basic protein (MBP)Ig-like domain-containing protein

A total of 45 proteins were identified as overlapping between the BT and SR groups. Notably, several of these shared proteins detected in both CSF and SR are implicated in critical disease-related pathways, including neurodegeneration, protein ubiquitination, and the regulation of cytoskeletal and synaptic function, as shown in Table 1.

### 3.3. Significant Pathways Identified from Functional Enrichment Analysis

Functional enrichment analysis of the significant candidate markers was performed using ShinyGO v0.81 (https://bioinformatics.sdstate.edu/go/ accessed on 23 February, 2025), with enrichment assessed at a false discovery rate (FDR) threshold of 0.05 using the KEGG pathways database. Enrichment analysis revealed no pathways in the SR proteins that were specifically relevant to rabies pathogenesis or associated clinical signs (Figure 6).

In contrast, analysis of the cerebrospinal fluid (CSF) candidate markers revealed significant enrichment in several disease-related pathways. Notably, these included spinocerebellar ataxia, neurodegeneration- multiple disease pathways, as illustrated in Figure 7.

The functions and features of proteins found in the spinocerebellar ataxia pathways are shown in Table 2.

Proteosomes found in the spinocerebellar ataxia pathway were also found in the pathway of neurodegenerative-multiple diseases and with the underlisted proteins, as shown in Table 3.

## 4. Discussion

This study reports on canonical protein signatures from brain tissue (BT), cerebrospinal fluid (CSF), and serum (SR) samples of dogs, which have the potential to be utilized in diagnosing rabies infection before clinical signs appear.

Leveraging the context of dog meat consumption in Nigeria, samples were collected from apparently healthy dogs slaughtered for human consumption. Lyssavirus antigens were detected in 20 out of 231 brain tissue samples tested.

In this study, we identified 54, 299, and 280 differentially abundant proteins in BT, CSF, and SR samples, respectively. The relatively low number of differentially abundant proteins identified in brain tissue may be attributed to the early stage of infection, as evidenced by the scanty distribution of lyssavirus antigen in the tested brain samples. This is further supported by the absence of clinical signs in the animals at the time of sampling, suggesting that the disease had not yet been fully established in the central nervous system.

One protein identified as overlapping between the CSF and BT datasets is Rho GDP-dissociation inhibitor beta (ARHGDIB), which plays a pivotal role in essential cellular processes, including actin cytoskeleton reorganization, cell adhesion, and cell motility. Notably, its increased abundance in the rabies-positive group has been associated with HIV replication and glioma progression [53,54]. Although direct evidence linking ARHGDIB abundance changes to rabies infection is unknown, its established role in regulating the actin cytoskeleton suggests potential involvement in viral entry, replication, and spread within host cells. As a critical regulator of Rho GTPases, which orchestrate cytoskeletal rearrangements, altered ARHGDIB activity may disrupt these processes, thereby influencing rabies virus infection dynamics and intercellular transmission.

Two proteins identified as overlapping between BT and SR were Myelin basic protein (MBP) and Ig-like domain-containing protein. MBP functions to promote myelin membrane adhesion, forming stable insulation around axons that enhances nerve impulse conduction speed. Increased abundance of myelin basic protein (MBP) is linked to multiple sclerosis and other CNS disorders, causing autoimmune demyelination, neuronal toxicity, and impaired myelin formation and maintenance [55]. In the context of RABV infection, myelin basic protein (MBP) has been implicated in neurological complications following rabies vaccination, particularly with older brain-derived vaccines. Studies have shown that immune responses triggered by vaccine components may generate antibodies against MBP, leading to autoimmune-mediated demyelinating disorders such as postvaccinal encephalomyelitis in a subset of recipients [56]. Schwann cells are responsible for the synthesis of the myelin sheath, which is composed primarily of myelin basic protein (MBP), as well as for its degradation during the demyelination process that follows peripheral nerve injury, such as that caused by a rabid animal bite. In this study, the observed increase in abundance of MBP may be attributable to the accumulation of rabies virus (RABV) within Schwann cells, which serve as target cell populations in peripheral nerves. Viral release from axons and infected Schwann cells represents a critical step in the RABV replication cycle and may contribute to RABV-induced demyelination of peripheral neurons and modulation of local innate immune responses, thereby facilitating viral persistence and pathogenesis [57].

According to the UniProt database [58], immunoglobulin-like domain-containing proteins currently lack assigned gene names and have not been characterized in canines. However, in humans, immunoglobulin-like domain-containing proteins defined by structural and sequence homology are found across a wide range of protein families. These domains are implicated in various biological functions, including cell–cell recognition, cell-surface receptor signalling, muscle organization, and immune system processes [59,60]. Studies have shown that viruses exploit Ig-like domain proteins to evade host immunity by suppressing or inhibiting immune responses, e.g., SARS-CoV-2 [41].

Among the 45 proteins that overlapped between CSF and SR were several proteins involved in key cellular processes, including the degradation of ubiquitinated proteins, cytoskeletal organization, and synaptic function. Notably, PDZ and LIM domain protein 1 (PDLIM1) is associated with cytoskeletal scaffolding and the regulation of synapse formation and maintenance, which are critical for neuronal communication [61]. Although this protein had not been previously studied in the context of rabies, its functional roles and observed differential abundance in this study suggest potential relevance in rabies pathogenesis. Rabies virus is known to enter peripheral neurons at axon terminals and relies on long-distance axonal transport and trans-synaptic spread to reach and infect the central nervous system (CNS) [57]. The altered abundance of proteins involved in synaptic integrity and cytoskeletal dynamics may facilitate the efficient neuronal transmission of the virus, contributing to the progression of infection.

Another noteworthy protein identified among the 45 that overlapped between CSF and SR is apolipoprotein C-IV (APOC4). While the role of APOC4 in rabies infection is not yet clearly defined, it is known to participate in lipid metabolism and may contribute to the regulation of triglyceride levels [62]. In contrast, apolipoprotein D (ApoD), a structurally related protein, has been shown to facilitate rabies virus (RABV) propagation by interacting with the viral glycoprotein (G protein) and upregulating cholesterol synthesis, an essential component for viral replication [24]. Apolipoprotein A1 has been proposed as a potential biomarker for the clinical diagnosis of rabies in virus-infected mice [23]. The identification and differential abundance of APOC4 in this study suggest a possible involvement in rabies pathogenesis. Further investigation into its function during infection may uncover novel mechanisms by which the virus interacts with host lipid regulatory pathways.

RNA transcription, translation, and transport factor protein (RTRAF) was identified among the 45 proteins overlapping between CSF and SR. RTRAF plays a pivotal role in gene expression by ensuring the accurate and efficient translation of genetic information into functional proteins. An increased abundance of this protein has been associated with neurodegenerative diseases, cancer, and developmental abnormalities [39]. This protein has not yet been described in the context of RABV infection; however, given its established role in neurodegeneration, its alteration in abundance may contribute to rabies pathogenesis, particularly through mechanisms linked to neuronal dysfunction.

Fold enrichment analysis in this study revealed two essential pathways: spinocerebellar ataxia and neurodegeneration-multiple diseases pathways that may be critically involved in the pathogenesis of rabies. Ataxia, characterized by impaired muscle coordination and unsteady gait, caused by viral damage of the cerebellum and brainstem, is a recognized clinical sign of rabies [63]. Among the most significantly enriched proteins within the spinocerebellar ataxia pathway were proteasome 26S subunit (PSMD1, PSMD7, PSMD12, PSMD13), specifically the non-ATPase regulatory components, which are central to the degradation of ubiquitinated proteins. Also, among the 45 proteins identified as overlapping between CSF and SR were notable proteasomes, which include the 26S proteasome non-ATPase regulatory subunit 2 (PSMD2), 26S proteasome non-ATPase regulatory subunit 3 (PSMD3), and the 26S proteasome regulatory subunit 10B (PSMC6). Previous studies have shown that certain RNA viruses can mimic the activity of E3 ubiquitin ligases, targeting host immune signalling proteins for ubiquitination and subsequent degradation, thereby subverting the host’s antiviral defense mechanisms [64,65]. In this study, the change in abundance of 26S proteasome subunits may not be entirely attributed to the presence of RABV in the host, as several other disease pathways identified in the fold-enrichment analysis could employ similar mechanisms to evade host immune responses. However, considering that RABV is an RNA virus, it is plausible that it utilizes a comparable strategy, which may account for the widespread change in abundance of proteasomal components observed. If RABV indeed exploits ubiquitin ligase mimicry to tag interferon pathway proteins for proteasomal degradation, this would provide a plausible mechanism for immune evasion during disease progression.

Solute Carrier Family 25 Member 6 (SLC25A6), one of the proteins identified in the spinocerebellar ataxia pathway, functions as a mitochondrial carrier protein that mediates ADP/ATP exchange across the inner mitochondrial membrane [48]. Altered abundance of SLC25A6 has been associated with ataxia, a neurological disorder characterized by impaired coordination, balance, and speech, which is also a recognized clinical manifestation of rabies. Because brain cells, particularly those within the cerebellum, have exceptionally high energy demands, disruption of ATP transport can result in cellular damage and neuronal death in regions responsible for motor coordination [66]. The Purkinje cells of the cerebellum, being large and metabolically active neurons critical for motor control, are especially susceptible to ATP deficiency, leading to premature degeneration. Studies have demonstrated that ATP in cerebellar nerve terminals is essential for maintaining ion gradients and facilitating neurotransmitter release; thus, reduced ATP availability can impair these functions, producing the motor dysfunctions characteristic of rabies infection [67]. Furthermore, impairment of mitochondrial function due to SLC25A6 alteration leads to decreased ATP production and increased oxidative stress, which in turn results in further neuronal injury [68]. This interplay between energy metabolism, oxidative stress, and neurodegeneration highlights the crucial role of SLC25A6 in the pathogenesis and progression of rabies.

In addition to the proteasome 26S non-ATPase subunits and SLC25A6 identified within the neurodegeneration-multiple diseases pathways, two other notable differentially abundant proteins, Ras-related C3 botulinum toxin substrate 1 (RAC1) and calcium/calmodulin-dependent protein kinase II beta (CAMK2B), were identified in this study and have previously been implicated in rabies pathogenesis. RAC1 has been reported in the brains of rabid dogs, and recent studies have shown that RABV infection induces cytoskeletal rearrangements and biphasic kinetics in RAC1 signalling [49]. Such cytoskeletal disruptions are strongly associated with neurological abnormalities and neurogenic disorders, leading to aberrant neuronal processes. The identification and differential abundance of RAC1 in this study suggest its critical role in the early stages of rabies pathogenesis.

CAMK2B, which mediates calcium/calmodulin-dependent protein kinase activity, is involved in regulating neuronal migration and synaptic signalling. While CAMK2B has not yet been studied in the context of rabies, its close homolog, CAMK2A, has previously been identified in rabid dog brains [22]. CAMK2A is well known for its role in calcium signalling, neuronal plasticity, and memory, and is highly enriched in the postsynaptic density (PSD), a key site for synaptic transmission. Given the functional similarities between CAMK2A and CAMK2B, the presence of CAMK2B in this study points to its potential involvement in rabies pathogenesis, particularly during the early phases of infection when neuronal communication is first disrupted.

Another vital protein identified in neurodegeneration-multiple disease pathways is PPP3R1, also known as protein phosphatase 3 regulatory subunit B, alpha isoform, which is a regulatory component of the calcineurin enzyme complex, playing a central role in calcium-mediated signal transduction [52]. Although PPP3R1 has not been directly linked to rabies infection, its established functions in synaptic transmission and neuronal signalling suggest that it may be indirectly impacted during the course of the disease. Given that the RABV primarily targets the nervous system and disrupts neuronal function, the alteration in the abundance of PPP3R1 and associated calcium signalling pathways could contribute to the development of neurological symptoms. Investigating the role of PPP3R1 in the context of rabies infection may provide valuable insights into the molecular mechanisms underlying rabies pathogenesis.

No significant pathways were identified during the fold enrichment analysis of brain tissue samples. This lack of enrichment may be attributed to the limited progression of disease within the brain at the time of sample collection. It is likely that rabies virus replication had not yet advanced sufficiently to trigger measurable alterations in major biological pathways. This is supported by the relatively small number of differentially abundant proteins identified in the brain tissue, suggesting that the infection may have been in its early stages and had not fully established within the central nervous system.

The infectious disease pathways identified in the fold enrichment analysis of SR samples included African trypanosomiasis, amoebiasis, Staphylococcus aureus infection, and prion disease. In Nigerian dogs, trypanosomiasis is primarily caused by Trypanosoma brucei and Trypanosoma congolense, both of which are haemoprotozoan parasites [69,70]. Amoebiasis, in contrast, is a gastrointestinal protozoan parasitic infection [71], while Staphylococcus aureus infection typically affects the skin and, in systemic cases, may involve multiple organs [72]. Prion disease is distinct from these conditions, as it results from the accumulation of misfolded prion proteins that induce neurodegeneration, rather than the infection and inflammation characteristic of rabies virus. Interestingly, dogs are naturally resistant to prion infections due to the presence of aspartic or glutamic acid at position 163 of their prion protein [73]. Therefore, the detection of the prion disease pathway in this study is unexpected and warrants further investigation.

## 5. Conclusions

This study identified candidate protein biomarkers in CSF and SR from apparently healthy, slaughtered dogs that may be indicative of early-stage rabies infection. These findings offer valuable insights into the molecular changes that occur prior to the onset of clinical symptoms and highlight the potential utility of these biomarkers in supporting early diagnosis. Ultimately, their application could enhance decision-making regarding the timely administration of post-exposure prophylaxis, thereby improving rabies prevention and control strategies.

## 6. Limitations of the Study

One key limitation of this study is the assumption that the sampled dogs were clinically healthy based solely on physical examination, as no physiological parameters or veterinary health records were available. This limitation stemmed from the fact that dog marketers who purchased the animals from their owners placed little emphasis on maintaining health documentation. Consequently, the assessment of the dogs was restricted to clinical observations and antemortem examination prior to slaughter. As a result, some animals may have been harboring other undiagnosed conditions. This is supported by the identification of disease-related pathways such as trypanosomiasis, amoebiasis, and systemic lupus erythematosus during the fold enrichment analysis of SR and CSF proteins. These co-existing conditions may have influenced protein abundance profiles and confounded the interpretation of rabies-specific biomarkers. For future studies, although a full veterinary evaluation of the animals may not be feasible due to the urgency of slaughter by dog marketers, systematic screening of samples to confirm that the animals are free from other diseases, apart from rabies, would help minimize potential confounders.

TRIzol and organic solvents were used to extract proteins from brain tissues and biofluids, respectively, as the samples were considered highly infectious. Since the infection status of the dogs was undetermined, all extractions were performed in a BSL-3 facility where no sonicator was available. Although a substantial number of proteins were identified, it is possible that the use of more advanced extraction methods could have yielded a greater number of detectable proteins, such as filter-aided sample preparation.

Furthermore, although relative protein quantification of canonical proteins is commonly employed in biomarker discovery studies, this limits insight into the biological functions of other proteoforms. Investigating the complexity that comes with the various modifications and conformations associated with proteins, which influence molecular function and thus pathophysiological conditions, may provide greater insight and precision in both diagnosis and prognosis, contributing to the identification of more specific and sensitive biomarkers for antemortem diagnosis of rabies [74]. In addition, future studies would focus on targeting the specific proteins identified in neurological pathways from the data generated in this discovery study, using the prm-PASEF method.

## Figures and Tables

**Figure 1 proteomes-13-00066-f001:**
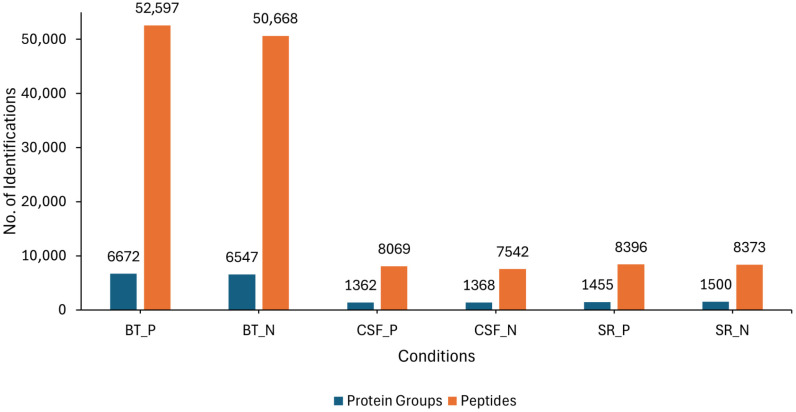
Showing the overall number of average protein groups and peptides obtained from the BT, SR and CSF positive and negative samples. The blue bars represent the identified protein groups, whereas the orange bars represent the identified peptides.

**Figure 2 proteomes-13-00066-f002:**
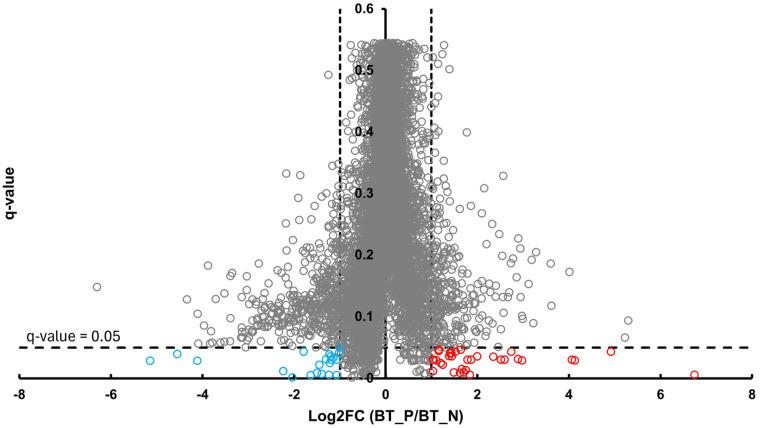
Volcano plot of BT illustrates an overview of the number of differentially abundant proteins identified. The red circles indicate the 33 proteins that had increased abundance, while the blue circles indicate the 21 proteins that had decreased abundance (*q*-value < 0.05, log_2_FC ≥ 1).

**Figure 3 proteomes-13-00066-f003:**
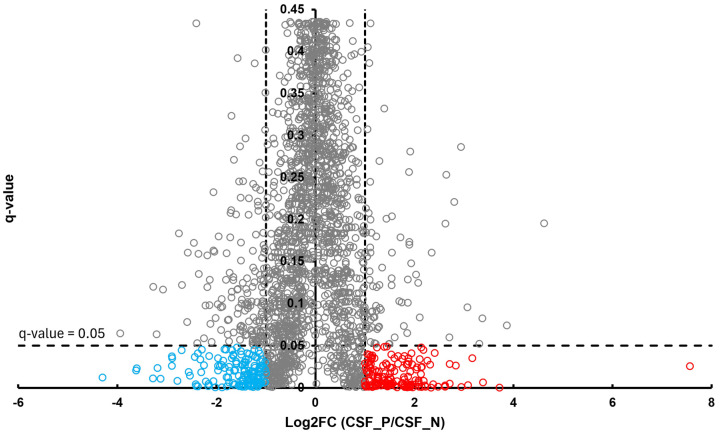
Volcano plot of CSF illustrates an overview of the number of differentially abundant proteins identified. The red circles denote the 159 proteins that had increased abundance, while the blue circles indicate the 140 proteins that had decreased abundance (*q*-value < 0.05, log_2_FC ≥ 1).

**Figure 4 proteomes-13-00066-f004:**
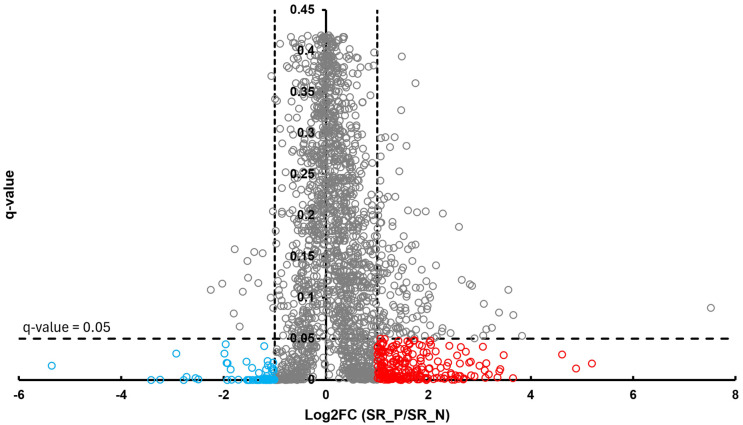
Volcano plot of SR illustrates an overview of the number of differentially abundant proteins identified. The red circles indicate the 215 proteins that had increased abundance, while the blue circles indicate the 65 proteins that had decreased abundance (*q*-value < 0.05, log_2_FC ≥ 1).

**Figure 5 proteomes-13-00066-f005:**
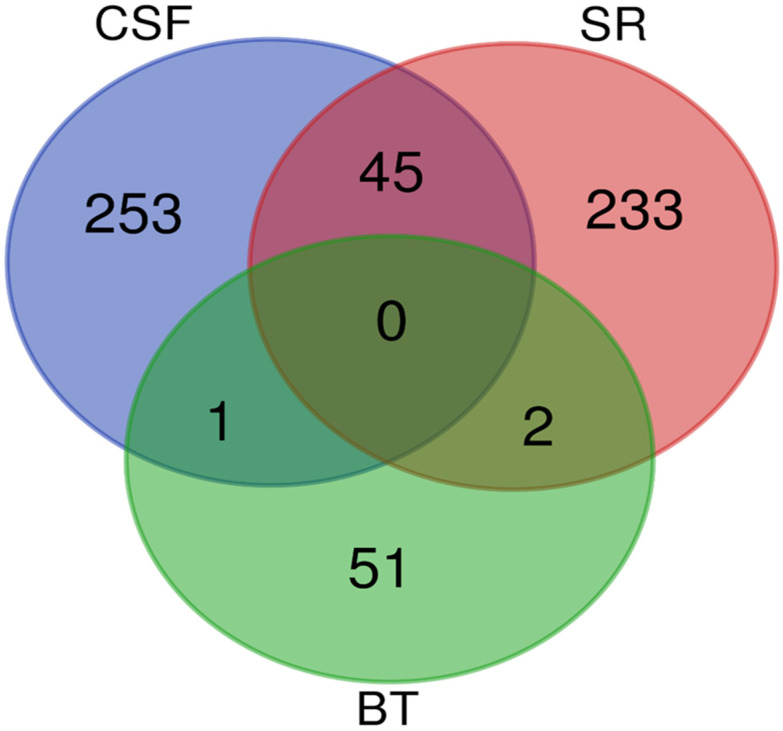
A Venn diagram showing the overlap of significantly differentially abundant proteins between the rabies-positive and -negative samples of the BT, CSF and SR groups. The green circle denotes proteins identified from brain tissues, the red circle denotes proteins identified from sera, and the blue circle denotes proteins identified from cerebrospinal fluid.

**Figure 6 proteomes-13-00066-f006:**
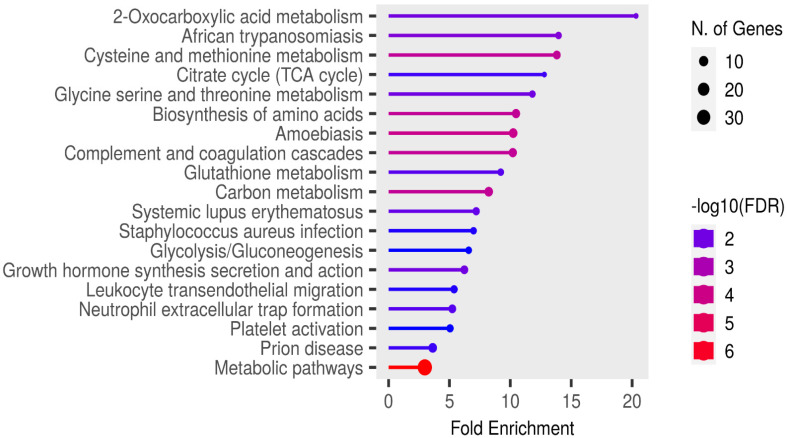
Functional enrichment analysis of the significant candidate markers found in sera using ShinyGo v0.81. Enriched KEGG pathways are shown ranked by descending order of fold enrichment (FDR of 0.05).

**Figure 7 proteomes-13-00066-f007:**
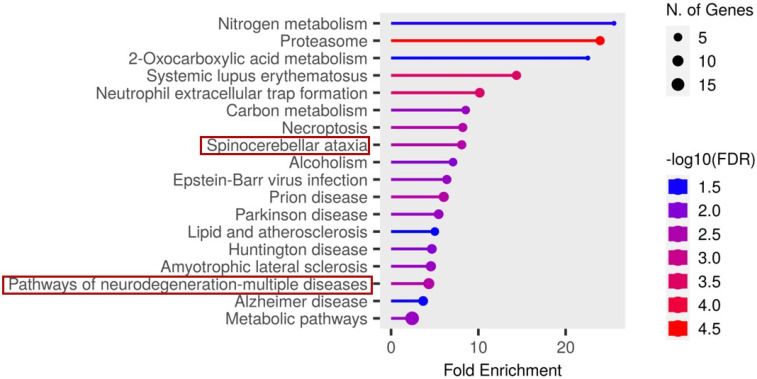
Functional enrichment analysis of the significant candidate markers found in CSF using ShinyGO v0.81. Enriched KEGG pathways are shown ranked by descending order of fold enrichment (FDR of 0.05).

**Table 1 proteomes-13-00066-t001:** Some important proteins among the 45 candidate markers that overlapped between CSF and SR. (See Appendix A for all the candidates).

SN	Protein	Gene	Function	Features
1	Proteasome 26S Subunit, ATPase 6	*PSMC6*	a subunit of the 26S proteasome, a crucial protein degradation complex. Ubiquitinated proteins are recognized, unfolded, and degraded by the proteasome.	As part of the proteasome, it can degrade proteins involved in activating the IFN pathway [34].
2	PDZ and LIM domain 1	*PDLIM1*	Cytoskeletal scaffold for assembling protein complexes. Supports synapse formation and maintenance for neuron communication.	Increased abundance in a variety of tumors and plays essential roles in tumor initiation and progression [35].
3	Apolipoprotein C-IV	*APOC4*	Plays a role in lipid metabolism, particularly related to triglyceride transport and clearance.	Increased abundance of the protein may influence circulating lipid levels and may be associated with coronary artery disease risk [36].
4	Nucleoside diphosphate kinase A	*NME1*	This enzyme maintains nucleotide homeostasis, supporting DNA/RNA synthesis, energy metabolism, and signal transduction.	Increase in abundance of nucleoside diphosphate kinases promotes neurite outgrowth and has been linked to lung tumor progression, while inactive forms suppress nerve growth factor activity [37].
5	26S proteasome non-ATPase regulatory subunit 3	*PSMD3*	It is involved in the ATP-dependent degradation of ubiquitinated proteins. participates in numerous cellular processes, including cell cycle progression, apoptosis, or DNA damage repair	Analysis revealed that an increase in abundance of PSMD3 is observed in multiple myeloma patients, with elevated levels significantly associated with poor patient survival [38].
6	RNA transcription, translation and transport factor protein	*RTRAF*	It is crucial for gene expression, ensuring precise and efficient translation of genetic information into functional proteins.	Altered abundance causes neurodegenerative disorders, cancer, and developmental abnormalities [39].
7	Triggering receptor expressed on myeloid cells 2	*TREM2*	The gene encodes a myeloid cell receptor vital for immune regulation, skeletal and neural development, and microglial functions such as inflammation, phagocytosis, and survival.	It has been implicated in neurodegenerative disorders such as Nasu-Hakola disease and Alzheimer’s disease, and may also contribute to Parkinson’s disease and amyotrophic lateral sclerosis [40].
8	Ig-like domain-containing protein	*LOC102724971*	Their primary role is molecular recognition and binding, supporting key processes such as cell–cell interactions, adhesion, and immune responses.	Viruses exploit Ig-like domain proteins to evade host immunity by suppressing or inhibiting immune responses, e.g., SARS-CoV-2 [41].
9	Vacuolar protein sorting-associated protein VTA1 homolog	*VTA1*	It plays a key role in the endosomal multivesicular body pathway, where it mediates the sorting of membrane proteins destined for degradation.	Altered abundance is linked to malignant choroidal melanoma and neurodegenerative conditions such as frontotemporal dementia and amyotrophic lateral sclerosis [42].
10	Glutamate--cysteine ligase	*GCLC*	It catalyzes the first step of glutathi one biosynthesis, joining L-glutamate and L-cysteine in an ATP-dependent reaction to form gamma-glutamylcysteine.	Reduced abundance has been linked to development of oxidative stress and schizophrenia [43].

**Table 2 proteomes-13-00066-t002:** Proteins differentially abundant in spinocerebellar ataxia pathway in clinical rabies.

Pathway	Protein (Gene Symbol)	Features
Spinocerebellar ataxia	Proteasome 26S Subunit, Non-ATPase 1 (PSMD1) (This protein is a subunit of the 26S proteasome, a large protein complex that breaks down ubiquitinated proteins, tagged for destruction).	Innate immune gene and cancer biomarker, including for oropharyngeal cancer, cystic fibrosis and Alzheimer’s disease [44].
	Proteasome 26S Subunit, Non-ATPase 7 (PSMD7)	Increase in abundance is linked to poor cancer prognosis; potential survival biomarker [45].
	Proteasome 26S Subunit, Non-ATPase 12 (PSMD12)	Altered abundance impairs protein degradation, contributing to neurodevelopmental disorders [46].
	Proteasome 26S Subunit, Non-ATPase 13 (PSMD13)	Increased abundance has been linked to endometrial cancer risk and treatment resistance in psychiatric disorders [47].
	Solute Carrier Family 25 Member 6 (SLC25A6) (Mitochondrial carrier protein mediating ADP/ATP exchange across the inner membrane)	Altered abundance activates inflammatory signalling pathways, resulting in the release of inflammatory cytokines, contributing to the progression of inflammation. It has been implicated in Alzheimer’s disease, Influenza and Bubonic Plague [48].

**Table 3 proteomes-13-00066-t003:** Proteins differentially abundant in pathway of neurodegenerative diseases-multiple diseases found in clinical rabies.

Pathway	Protein	Features
Neurodegenerative—multiple diseases	Ras-related C3 botulinum toxin substrate 1 (RAC1): A Rho-GTPase involved in cytoskeletal remodeling and survival.	Closely associated with neuronal dysfunction. RABV infection led to the rearrangement of the cytoskeleton as well as the biphasic kinetics of the Rac1 signal transduction, leading to neurological disorder [49].
	Glutathione Peroxidase 1 (GPX1): A key antioxidant enzyme that helps protect cells from the damaging effects of reactive oxygen species (ROS)	GPX1 is increasingly abundant in most human cancers, e.g., Kidney renal papillary cell carcinoma [50].
	Calcium/calmodulin-dependent protein kinase II B (CAMK2B): Function in long-term potentiation and neurotransmitter release essential for learning and memory	Activity is altered in Alzheimer’s disease, epilepsy, and ischaemic stroke [51].
	Protein phosphatase 3, regulatory subunit B, alpha isoform (PPP3R1). It regulates neuronal calcium signalling, synaptic transmission, receptor internalization, and the synaptic vesicle cycle.	It is associated with dilated cardiomyopathy, schizophrenia, and has also been implicated in Alzheimer’s disease [52].

## Data Availability

The mass spectrometry proteomics (raw) data along with the output file from SpectronautTM 19 have been deposited to the ProteomeXchange Consortium via the PRIDE [109] partner repository with the dataset identifier: PXD070262.

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
