# Peer review of "Proteome Profiling of Rabies-Infected and Uninfected Dog Brain Tissues, Cerebrospinal Fluids and Serum Samples"

_proteomes, 2025, doi:10.3390/proteomes13040066_

Round 1
Reviewer 1 Report
Comments and Suggestions for Authors
The manuscript prepared by Ukamaka Uchenna Eze et. al. describes a survey proteomics analysis of rabies-infected dogs. The authors analyzed brain tissues, CSF, and serum samples using DIA mass spectrometry, and differentially expressed proteins were identified. The article is interesting and well written; however, there are several important issues that have to be addressed before publication.
1, In case of all mass spectrometry, it is mandatory to make the raw data publicly available in online databases, such as ProteomeXchange, MassIVE etc. Without publicly available data, the reviewers cannot assess the data quality or the method of interpretation.
2, A more detailed description of the applied mass spectrometry method is needed in the method section.
3, Please discuss why only 20 samples out of 40 prepared samples were analyzed. What were the applied selection criteria?
4, The volcano plots show log10 p-values but the authors described FDR-corrected q values. In this case, the volcano plots should indicate the adjusted p-values.
5, The interpretation of the GO enrichment analysis needs to be clarified. In the case of Figure 6, the authors mention that “Enrichment analysis revealed no pathways in the SR proteins that were specifically relevant to rabies pathogenesis or associated clinical signs.” The enriched GO terms clearly indicate the presence of infection.
After the necessary changes related to the manuscript and data availability, the manuscript can be considered for acceptance.
Author Response
Dear Aubrey,
We would like to thank you for facilitating the review of our manuscript on “Proteome profiling of rabies-infected and uninfected dog brain tissues, cerebrospinal fluids and serum samples”
We have now attended to all the comments raised by the reviewer:
Reviewer 1:
The manuscript prepared by Ukamaka Uchenna Eze et. al. describes a survey proteomics analysis of rabies-infected dogs. The authors analyzed brain tissues, CSF, and serum samples using DIA mass spectrometry, and differentially expressed proteins were identified. The article is interesting and well written; however, there are several important issues that have to be addressed before publication.
Response: The authors thank the reviewer for the prompt and thoughtful review of our manuscript.
1, In case of all mass spectrometry, it is mandatory to make the raw data publicly available in online databases, such as ProteomeXchange, MassIVE etc. Without publicly available data, the reviewers cannot assess the data quality or the method of interpretation.
Response: The raw data, sample metadata, as well as the search engine output file have been submitted to ProteomeXchange and the data availability statement with the identifier number has been added to the manuscript as required.
Reviewer access details
Log in to the PRIDE website using the following details:
Project accession: PXD070262
Token: WG85uBht7xk8
Alternatively, reviewer can access the dataset by logging in to the PRIDE website using the following account details:
Username: reviewer_pxd070262@ebi.ac.uk
Password: A52MOwYehiWg
2, A more detailed description of the applied mass spectrometry method is needed in the method section.
Response: Additional details have been added to the description of the applied mass spectrometry method and can be found in from line 202 – 215.
3, Please discuss why only 20 samples out of 40 prepared samples were analyzed. What were the applied selection criteria?
Response: 40 samples from each sample type underwent protein extraction as described in 2.5) of the methods section. Following total protein quantification, only samples containing at least 20 µg of protein were selected for further sample preparation, as this was the minimum amount of protein to ensure that there would be sufficient for LC-MS/MS analysis after sample clean-up. The cerebrospinal fluid (CSF) samples had the least total protein content and a total of 10 negative, and 14 positive CSF samples met this criterion; however, to ensure uniformity across groups, 10 rabies-infected and -uninfected samples from each sample type were utilized for downstream analyses. See lines 175 – 183.
4, The volcano plots show log10 p-values, but the authors described FDR-corrected q values. In this case, the volcano plots should indicate the adjusted p-values.
Response: The volcano plots have been adjusted to indicate q-values on the y-axis as requested. See pages 7 – 10.
5, The interpretation of the GO enrichment analysis needs to be clarified. In the case of Figure 6, the authors mention that “Enrichment analysis revealed no pathways in the SR proteins that were specifically relevant to rabies pathogenesis or associated clinical signs.” The enriched GO terms clearly indicate the presence of infection.
Response: The infections identified in the GO enrichment analysis of SR proteins included African trypanosomiasis, amoebiasis, Staphylococcus aureus infection, and prion disease. In Nigerian dogs, trypanosomiasis is primarily caused by Trypanosoma brucei and Trypanosoma congolense, which are haemoprotozoan parasites. Amoebiasis, on the other hand, is a gastrointestinal protozoan parasitic infection, while Staphylococcus aureus infection typically affects the skin and, in systemic cases, may involve multiple organs. Prion disease is distinct among these conditions, as it is caused by misfolded proteins that affect the brain, leading to neurodegeneration rather than the infection and inflammation characteristic of rabies virus. See lines 505-515.
Once again, we would like to thank you for facilitating this review process.
Sincerely,
Kind regards,
Eze, U.U.
Reviewer 2 Report
Comments and Suggestions for Authors
Please find attached file for comments and reviews.
Best.

Author Response
Dear Aubrey,
We would like to thank you for facilitating the review of our manuscript on “Proteome profiling of rabies-infected and uninfected dog brain tissues, cerebrospinal fluids and serum samples”
We have now attended to all the comments raised by the reviewer:
Reviewer 2
The manuscript by Ukamaka et al. investigates the protein dysregulation between healthy and rabies-infected tissue samples from dogs, aiming to identify potential biomarkers for early detection of rabies. The authors examined three biological sample types, including brain tissue, cerebrospinal fluid and sera, and reported a substantial number of differentially expressed proteins across these matrices. Through proteomic profiling, the authors highlight several upregulated proteins as potential diagnostic indicators of rabies. While the dataset provides informative comparison between samples, additional experimental control and clarification on some of the claims in discussion section are necessary before concluding that these findings are specific to rabies infection.
Response: We thank the reviewer for the timely and constructive revision of our manuscript.
Major points
- Specificity of identified biomarkers:
The primary concern relates to the selection of candidate proteins presented in Table 1 and in discussion as potential biomarkers. Many of the dysregulated (both up- and downregulated) are commonly associated with general viral infection rather than rabies-specific. The authors discussed this in lines 407-414, noting that involvement of certain pathways are hallmark of RNA viruses. Indeed, many of these pathways including cellular recycling, ubiquitination and mitochondrial metabolism are frequently hijacked by RNa viruses, particularly respiratory types. This raises a critical question whether the observed proteomic changes represent rabies-specific signature, or do they reflect generalized antiviral response? The pathway that is distinctive to rabies pathology are those discussed in 397-406 relating spinocerebellar ataxia and neurodegenerative disease-associated proteins. The authors are highly encouraged to better delineate these findings and emphasize by having deep-dive into few selective rabies-specific hits to strengthen the paper.
Response: We have highlighted proteins that are specific to spinocerebellar ataxia, while others have been previously discussed, particularly those identified within the neurodegenerative pathway (Lines 436–465).
The reference used from Javier et al., 1989 appears to be overstated. The study reported post-vaccinal EAE due to the presence of MBP in older vaccine formulations. It seems like an over assumption as MBP is not causal nor result from active rabies infection, but rather a component of older generation vaccine preparation. On other note, MBP is a well-established marker of Schwann cells in the brain. Notably prior studies have demonstrated that the RABV P protein accumulates in the cytoplasm of Schwann cells in mice model. Could this be an alternative and potential dysregulation of MBP that is directly downstream effect of RABV infection, which could account for the distinctive biomarker pattern observed? Key reference: o Potratz etal., 2020. Acta Neuropathol Comm. 8:199. doi: 10.1186/s40478-020-01074-6
Response: We thank the reviewer for highlighting the importance of Schwann cells in rabies pathogenesis. We also appreciate the suggested reference, which has been incorporated into the manuscript along with relevant comments discussing the effect of MBP dysregulation on rabies pathogenesis (Lines 374–382).
- The proteomic hits identified in sera (figure 6) are predominantly related to metabolism. This outcome may be expected, as many disease- or virus-related proteins detectable in serum typically reflect cellular stress or apoptosis. Have the authors tried quantifying other circulating factors such as immune cells, or secreted immunoglobulins? (IgM, IgG, IgE?)
Response: No, we did not identify any specific immunoglobulins in this study. Instead, we detected immunoglobulin-like domain-containing and immunoglobulin domain-containing proteins. Therefore, circulating immune cells were not quantified
- Figures 6 and 7 present fold enrichment analyses highlighting potential biomarker candidates. However, the discussion primarily focuses on upregulated proteins. Were there any cluster or families of downregulated proteins that could suggest functional suppression or pathological alteration associated with rabies infection?
Response: All the significantly differentially abundant proteins identified in each group (i.e., BT, CSF, and Sr) were used to generate figures 6 and 7; these are proteins that showed increased and decreased abundance in each group. When the functional enrichment analysis was performed on ShinyGO (under the same conditions used to generate figure 6 and 7) using the KEGG database specifically for the candidate proteins with decreased abundance in the groups, no significantly enriched pathways were given.
Minor/Miscellaneous points 1. Consistency on formatting the authors. E
Some were noted et al. some were noted et al. o Line 160 - Nejadie et al. needs to be italicized
Response: it has been corrected
Font/Size inconsistency o Line 263, 268 273 = different font/size was used for FC=1
Response: It has been corrected
Line 296 - Improper capitalization for “Important”
Response: It has been corrected
- Possible type error e Line 34 -SrRgroup should be SR group
Response: It has been corrected
Once again, we would like to thank you for facilitating this review process.
Sincerely,
Kind regards,
Eze, U.U.
Round 2
Reviewer 1 Report
Comments and Suggestions for Authors
The authors answered all of my questions and made the necessary changes. The manuscript can be accepted for publication.
Author Response
Comment: The authors answered all of my questions and made the necessary changes. The manuscript can be accepted for publication.
Response: We appreciate Reviewer 1’s positive recommendation for the publication of our manuscript, following our satisfactory completion of all requested revisions.

Reviewer 2 Report
Comments and Suggestions for Authors
The authors have addressed all minor concerns, and have partially addressed major concerns. The manuscript would have been substantially stronger with the addition and/or comparison with other viruses, nonetheless the addition of tables on RABV clinical data relevance did enhance the work and supports their hypothesis that many of the observed results are closely associated to RABV-specific infection.
A few minor revisions still required. In particular the volcano plots legend and especially Figure 3 appears to have been placed on top of a previous version of the figure, as remnants of the earlier legend and y-axis are still visible to me in pdf format. This, alongside with few editorial edits (double spaced in some cases, legends being vertical etc) should be corrected to ensure clarity in the final manuscript presentation.
Author Response
comments: A few minor revisions still required. In particular the volcano plots legend
and especially Figure 3 appears to have been placed on top of a previous
version of the figure, as remnants of the earlier legend and y-axis are still
visible to me in pdf format. This, alongside with few editorial edits (double
spaced in some cases, legends being vertical etc) should be corrected to
ensure clarity in the final manuscript presentation.
Response: We have removed all previous volcano plots, and only the updated versions are now included. We have also eliminated all double spacing. However, we did not identify any vertical legends in the manuscript. We kindly request the reviewer to specify the exact location of the vertical legends so that we may address them appropriately.